



# Interaction of Small-Scale Gravity Waves with the Terdiurnal Solar Tide in the Mesosphere and Lower Thermosphere

Friederike Lilienthal[1], Erdal Yiğit[2], Nadja Samtleben[1], and Christoph Jacobi[1]

[1]Institute for Meteorology, Universität Leipzig, Stephanstr. 3, 04103 Leipzig, Germany
[2]Department of Physics and Astronomy, Space Weather Lab, George Mason University, Fairfax, VA, USA

**Correspondence:** Friederike Lilienthal (friederike.lilienthal@uni-leipzig.de)

**Abstract.** Implementing a nonlinear whole atmosphere gravity wave (GW) parameterization into the Middle and Upper Atmosphere Model extending to the lower thermosphere (160 km), we study the response of the atmosphere in terms of the circulation patterns, temperature distribution, and migrating terdiurnal solar tide activity to the upward propagating small-scale internal GWs originating in the lower atmosphere. We perform three test simulations for the Northern Hemisphere winter conditions in order to assess the effects of variations in the initial GW spectrum on the dynamics of the mesosphere and lower thermosphere. We find that the overall strength of the source level momentum flux has a comparatively small impact on zonal mean dynamics. The tails of the GW source level spectrum, however, are crucial for the lower thermosphere dynamics. With respect to the terdiurnal tide, we find a strong dependence of tidal amplitude on the induced GW drag, generally being larger when GW drag is increased.

## 1 Introduction

Atmospheric gravity waves (GW) are known to cause a variety of effects in the middle and upper atmospheres of all planetary atmospheres that have been studied so far (Yiğit and Medvedev, 2019). Historically, the importance of GWs for atmospheric dynamics has been acknowledged first in the context of Earth's middle atmosphere (e.g., Holton, 1982). For about two decades later, it has been widely assumed that small-scale GW effects are confined to the mesosphere. However, a number of studies, especially since the second half of 2000s, have shown that GW effects extend well into the thermosphere (e.g., Yiğit et al., 2009; Hickey et al., 2009; Yiğit and Medvedev, 2010; Heale et al., 2014; Gavrilov and Kshevetskii, 2015), while coordinated observations also demonstrate thermospheric GW signatures (e.g., Forbes et al., 2016; Trinh et al., 2018) that cannot be explained by considering solely solar and magnetic effects. Meanwhile, GWs are acknowledged as an important mechanism that contributes to the vertical coupling in the atmosphere-ionosphere system as has been discussed in contemporary reviews (e.g., Yiğit and Medvedev, 2015). During transient events such as sudden stratospheric warmings, thermospheric effects of GWs can be extremely variable depending on the nature of the warming (Yiğit and Medvedev, 2016; Nayak and Yiğit, 2019). Often, simple linear-type GW parameterizations with ad hoc cut-off levels in the upper mesosphere and lower thermosphere have been used in order to represent small-scale GWs not captured in coarse-grid general circulation models (GCMs). However, recent





progress in GW dynamics suggests that GW schemes based on more accurate GW physics are required in order to adequately
represent subgrid-scale GW processes in GCMs (Yiğit et al., 2008).

A broad spectrum of internal waves exists in the atmosphere. While GWs have relatively small scales with respect to the planetary radius, solar tides are large-scale waves with horizontal wavelengths comparable to Earth's radius. The most predominant types of atmospheric tides are the migrating diurnal (DTs), semidiurnal (SDTs), and the terdiurnal tides (TDTs). Despite the large differences in scales between the GWs and tides, they continuously interact with each other, potentially
producing secondary waves, which can then influence the upper atmosphere (Forbes et al., 1991; Miyahara and Forbes, 1991; Manson et al., 2002; Yiğit and Medvedev, 2017; Lilienthal et al., 2018; Lilienthal and Jacobi, 2019). However, owing largely to the complexity of the interaction processes, there is an ongoing discussion about how GWs influence the solar tides. While significant amount of work has been conducted on the relation between GWs, DTs, and SDTs, the progress on the understanding of the interaction between GWs and the TDTs is relatively limited. Also, the vast majority of the studies focuses on the MLT
region in the context of GW-tide interactions. For example, using a numerical model of the DT coupled with simplified linear GW drag calculations, including only slow GW phase speeds, Miyahara and Forbes (1991) demonstrated that GW drag damps the tidal amplitudes in the MLT. The study by Manson et al. (2002), combining observations and a GCM, suggested that the tidal response highly depends on the type of the utilized GW parameterization. Further model simulations by Lilienthal et al. (2018) and Lilienthal and Jacobi (2019) have shown that the GW-tide interactions can generate TDTs that can particularly
be important for the dynamics of the lower thermosphere. Overall, existing results on the GW-tide interactions all suggest that there is a distinct difference between linear and nonlinear GW schemes in terms of how they influence the solar tides. Undoubtedly, linear GW schemes provide only a limited picture of the actual GW dynamics in the atmosphere.

Our study is motivated by recent progress in GW studies and the lack of knowledge concerning the nature of GW-tide interactions. Specifically, we implement for the first time a state-of-the-art nonlinear whole atmosphere GW parameterization
(Yiğit et al., 2008) into the Middle and Upper Atmosphere Model (MUAM) used at the University of Leipzig, Germany. We then study the interaction between the TDTs and GWs accounting for lower thermospheric GW effects in addition to the middle atmospheric effects.

The structure of the paper is as follows: Next section describes in detail the GCM, GW parameterization, and the simulations to be conducted. Section 3 presents the simulation results for the zonal mean fields based on the standard configuration of the
GW scheme; section 4 studies the effects of changing initial GW parameters; and section 5 analyzes the interaction between GWs and the migrating TDT. Summary and conclusions are given in section 6.

## 2 Model Description, Gravity Wave parameterization, and Experimental Setup

In the following experiments, we use the Middle and Upper Atmosphere Model (MUAM; Pogoreltsev, 2007; Pogoreltsev et al., 2007; Suvorova and Pogoreltsev, 2011), which is a three-dimensional mechanistic GCM solving the nonlinear primitive
equations (e.g., Jakobs et al., 1986). The 1000 hPa layer is the lower boundary of MUAM based on 2000-2010 mean monthly mean ERA-Interim reanalysis fields (Dee et al., 2011) of zonal mean temperature and geopotential height as well as the





respective stationary planetary waves (SPWs) with wavenumbers 1-3 (see also Lilienthal et al., 2017). The horizontal resolution is $5° \times 5.625°$ (latitudes $\times$ longitudes). There are 56 vertical levels, which are evenly spaced in logarithmic pressure height with $p_s = 1000$ hPa as the reference pressure level and $H = 7$ km as the scale height. Vertical spacing is about 2.8 km, and

consequently, the upper boundary is located at $z_{56} \approx 160$ km. The lower levels up to 30 km height are nudged with 2000-2010 mean monthly mean ERA-Interim zonal mean temperature fields to correctly represent the dynamics in the lower atmosphere (Jacobi et al., 2015; Lilienthal et al., 2018). Solar and infrared radiative processes are parameterized according to the works by Strobel (1978) and Fomichev and Shved (1985), respectively. These parameterizations focus on (i) the absorption and emission processes of the most important atmospheric constituents like $H_2O$ (troposphere), $CO_2$ and $O_3$ (stratosphere) as well as on

(ii) absorption bands like the extrem ultra violet (EUV) band in the thermosphere. The $H_2O$, $CO_2$, and $O_3$ distributions are prescribed. Further parameterizations deal with thermospheric processes such as Rayleigh friction, ion drag, and Newtonian cooling.

Solar tides are generated self-consistently in the model by the absorption of solar radiation, mainly due to water vapor and ozone. Unlike other mechanistic models, there is no explicit tidal forcing at the lower boundary. The sources of TDTs

within MUAM were demonstrated in the work by Lilienthal et al. (2018). These sources are predominantly solar heating in the troposphere and stratosphere, nonlinear interactions between the DT and SDT in the mesosphere, and GW-tide interactions in the thermosphere.

In contrast to earlier MUAM versions (e.g., Jacobi et al., 2006), where a linear, Lindzen-type GW scheme was applied, we now use a nonlinear whole atmosphere GW scheme according to the work by Yiğit et al. (2008). An increasing number of whole

atmosphere models are being developed around the world, which further indicate the necessity of the use of GW schemes that are suitable for the whole atmosphere region. The whole atmosphere scheme describes the upward propagation of small-scale GWs and their dissipation due to various realistic atmospheric dissipation processes, such as nonlinear interactions, molecular diffusion and thermal conduction, turbulent viscosity, ion drag, and radiative damping. The most dominant dissipative processes in the middle and upper atmosphere are the nonlinear interactions between GWs and the dissipation due molecular processes.

This physical rationale and the detailed description of the scheme are given in a number of publications (e.g., Yiğit et al., 2008; Yiğit and Medvedev, 2013; Miyoshi and Yiğit, 2019). It has been also used in studies of GW effects with Martian GCMs (e.g., Yiğit et al., 2018).

Vertical profiles of the Newtonian cooling coefficient, the eddy diffusion coefficient and electron density are currently not used in our implementation of the parameterization, i.e. these parameters are set to zero. The source level of GWs is defined at

about 15 km similar to previous implementations of the whole atmosphere scheme (e.g., Yiğit et al., 2014; Yiğit and Medvedev, 2017). The GW spectrum specifies the GW momentum fluxes as a function of horizontal phase speeds.

We first conduct a reference (benchmark) simulation, which will later facilitate an assessment of the changes induced by variations in the GW source spectrum. The benchmark case (called EXP1 hereafter) is generated by spinning up the model for a period of 390 days, in which the mean circulation is built up and different waves such as GWs (after day 60), planetary waves

and tides (after day 180) are included. After the spin-up, we run the model for 30 days, which are used for our analysis of the background dynamics (zonal/meridional wind and temperature) and wave parameters shown in sections 3-5. Because MUAM





is driven only by monthly mean boundary conditions and reaches almost a steady state with small day-to-day variations after the spin-up period, the average of these last 30 days represents the monthly mean state of the atmosphere.

For the GW spectrum of EXP1, we adapt the original spectrum by Yiğit et al. (2008). It includes a total of $nh = 30$ harmonics
with the horizontal phase speeds $c_i$ ranging between $\pm 2$ and $\pm 80\,\mathrm{m\,s^{-1}}$. The peak momentum flux at the source level is $\overline{u'w'}(z_0) = 0.00025\,\mathrm{m^2\,s^{-2}}$ and the full width at half maximum (FWHM) of the spectrum function is located at $c_w = 35\,\mathrm{m\,s^{-1}}$. The horizontal wavelength of GWs is assumed to be $\lambda_H = 300\,\mathrm{km}$. This spectrum has also been used in a number of recent publications (e.g., Yiğit et al., 2009, 2012; Miyoshi and Yiğit, 2019).

In an additional experiment (EXP2) we retained the properties of the GW spectrum, except for increasing the peak mo-
mentum flux at the source level to $\overline{u'w'}(z_0) = 0.00035\,\mathrm{m^2\,s^{-2}}$. For a third experiment (EXP3), the peak momentum flux was the same like in EXP1, but the spectrum was modified now including $nh = 34$ harmonics and a FWHM of $c_w = 26\,\mathrm{m\,s^{-1}}$. Thereby, the total momentum flux at the source level, i.e. $\sum_i \overline{u'w'}_i(z_0)$ is the same for EXP3 like for EXP1. The GW spectra for EXP1, EXP2, and EXP3 are presented in Fig. 1.

## 3 Background Circulation

We first study the results of the benchmark simulation (EXP1) based on the standard GW spectrum. Figure 2 shows altitude-latitude cross sections of the monthly mean zonal mean (a) zonal ($u$) and (b) meridional wind ($v$) as well as the (c) neutral temperature ($T$) for Northern Hemisphere (NH) winter conditions (January). Figure 3 shows the GW effects in the same manner for (a) zonal GW drag, (b) meridional GW drag, and (c) GW heating/cooling. A strong westerly wind system exceeding $80\,\mathrm{m\,s^{-1}}$ is prevalent in the middle atmosphere in the NH, while summer easterlies dominate the Southern Hemisphere (SH).
These middle atmosphere jets extend up to an altitude of about $90\,\mathrm{km}$ (Fig. 2a), and significantly influence the upward propagation condition of small-scale GWs via wave filtering and critical level interactions. Thus, in the NH (winter) mainly westward directed GWs can propagate into the upper atmosphere, while the eastward directed GWs are substantially damped or largely filtered out. The opposite phenomenon prevails in the summer Southern Hemisphere (SH). This distribution of GW drag is clearly seen by the eastward GW drag in the SH and westward GW drag in the NH between $80$-$100\,\mathrm{km}$ (Fig. 3a), which is
primarily responsible for the reversal of the zonal mean flow above $90\,\mathrm{km}$, from eastward to westward direction ($\sim -40\,\mathrm{m\,s^{-1}}$) in the winter NH and from westward to eastward direction in the SH ($\sim 50\,\mathrm{m\,s^{-1}}$). The GW drag with alternating sign above $100\,\mathrm{km}$, for example, westward and eastward GW drag regime around $105$-$125\,\mathrm{km}$ in the SH and NH, respectively, is formed by the faster GW harmonics that have survived the filtering and nonlinear dissipation in the mesosphere (Yiğit et al., 2009). Owing to nonlinear interactions and increasing dissipation with altitude due to molecular diffusion and thermal conduction,
the surviving faster GW harmonics are attenuated in the thermosphere and produce drag there. Overall, maximum GW drag is in the order of $\pm 100\,\mathrm{m\,s^{-1}\,d^{-1}}$ and is stronger in the summer SH than the winter NH, which is in line with radar observations of GW fluxes and variance (Placke et al., 2011b, a).

The mean meridional circulation (Fig. 2b) is directed from the summer mesopause to the winter mesopause and has a maximum of $8\,\mathrm{m\,s^{-1}}$ around $80\,\mathrm{km}$. This is the upper branch of the so called Brewer-Dobsen Circulation. Like the zonal wind,





the meridional winds also reverse the direction, e.g., winter to summer mean flow around $110\,\mathrm{km}$. Overall, these circulation cells leading to the reversals in the zonal and meridional winds in the MLT are driven by GW dynamics. The associated changes in the residual mean circulation lead to the adiabatic cooling and warming of the summer and winter hemispheres, respectively, as can be seen by the temperature distribution in the MLT (Fig. 2c). Owing to the strong northward winds at around $80\,\mathrm{km}$, air descends and warms up the winter mesopause adiabatically ($\sim 180\,\mathrm{K}$), while it ascends and cools down adiabatically the

summer mesopause ($< 150\,\mathrm{K}$). During polar night, when the polar vortex establishes, there is a lack of incoming solar radiation so that the temperature is decreasing in the NH in the polar stratosphere. However, in the SH, the temperature rises up to $270\,\mathrm{K}$ in the polar stratosphere due to the absorption of solar radiation by ozone. Above $120\,\mathrm{km}$ in the thermosphere the temperature gradually increases, exceeding, e.g., $900\,\mathrm{K}$ in summer due primarily to the enhanced absorption of solar UV and EUV by molecular and atomic oxygen. GW heating and cooling [in $\mathrm{K}\ \mathrm{d}^{-1}$] presented in Fig. 3c suggests that the thermal effects of

GWs increase with increasing altitude in the thermosphere. The primary thermal effect of GWs is to cool the thermosphere above $120\,\mathrm{km}$ altitude, which is in agreement with previous studies (Yiğit and Medvedev, 2009; Yiğit and Medvedev, 2017). The thermal effects of GWs are produced by the combination of the frictional heating and the differential (dynamic) cooling by dissipating GWs (Medvedev and Klaassen, 2003; Yiğit et al., 2008).

We compare our results with reference climatologies such as the Upper Atmosphere Research Satellite (UARS) Reference

Atmosphere Project (URAP; Swinbank and Ortland, 2003) or the Committee on Space Research (COSPAR) International Reference Atmosphere (CIRA-86; Fleming et al., 1988) and with the more modern Global Empirical Wind Model (GEWM; Portnyagin et al., 2004; Jacobi et al., 2009) and the Horizontal Wind Model (HWM-14; Drob et al., 2015). The overall mean structure of the middle atmosphere and lower thermosphere are well reproduced by the model except for (i) the slightly overestimated mesospheric jet at $60\,\mathrm{km}$ between $50°\mathrm{N}$ and $65°\mathrm{N}$, which reaches up to $80\,\mathrm{m\,s}^{-1}$, and (ii) the missing tilt of

the mesospheric jet towards lower latitudes with increasing height. Samtleben et al. (2019) already reported a relatively large mesospheric jet in the MUAM model based on a tuned linear Lindzen-type GW parameterization, which was about $60\,\mathrm{m\,s}^{-1}$, being comparable to CIRA-86 and HWM-14, but about $20\,\mathrm{m\,s}^{-1}$ larger than the jet maximum proposed by GEWM and URAP. Note, however, that GEWM is only available for altitudes between $70$ and $100\,\mathrm{km}$ (slightly above our jet maximum) and URAP is interpolated for large areas near the jets. Due to the GW scheme after Yiğit et al. (2008), included in the present

simulations, the mesospheric jet has strengthened by additional $20\,\mathrm{m\,s}^{-1}$ and the jet maximum is slightly shifted towards the North compared to the MUAM simulations presented in the work by Samtleben et al. (2019). The zonal wind jet is of similar magnitude as the one simulated by Miyoshi and Yiğit (2019), but again stronger than the one predicted by the WACCM-X model (Qian et al., 2019). As in WACCM-X, the magnitude and altitude–latitude structure of the meridional MLT wind jet in MUAM is comparable to the ones predicted by the radar-based GEWM (Portnyagin et al., 2004; Jacobi et al., 2009). Our

results can also be compared with meteor radar retrievals of mesospheric winds in the $\sim 85\text{-}90\,\mathrm{km}$ range. For example, in the zonal wind climatology determined by Pramitha et al. (2019) it is seen overall that weak eastward winds reverse direction to westward at higher altitudes during January at equatorial and low-latitudes in the NH, which is also seen in our simulations.





## 4    Effect of modification of Gravity Wave Parameters

GW generation processes in the lower atmosphere are complex and a number of processes contribute to the formation of
the GW spectrum. We next would like to test the response of the GCM to variations in the initial GW spectrum. Two more
experiments have been performed as has been described in section 2. The same mean fields are analyzed and presented for these
simulations as has been done for EXP1. The field of a respective experiment is shown as contour lines, while the differences
with respect the benchmark run (i.e., EXP2-EXP1 and EXP3-EXP1) are given in color shading.

### 4.1    Modified GW Spectrum: Increased Flux at the Source Level

The impact of an increased GW flux at the source level on the mean circulation is shown in the difference plots in Fig. 4.
Fig. 4a indicates that right below the region of eastward wind reversal in the SH and around the westward wind reversal in the
NH, the mean zonal wind has become relatively westerly and easterly, respectively. Studying the associated GW drag results
in Figure 5a can provide some insight into this result. Increasing the source maximum momentum strength affects primarily
the dissipation altitude, thus the saturation level of individual GW harmonics as well as the associated drag produced by them.
Larger momentum flux means that GWs dissipate at lower altitude due to increased nonlinear interactions in the MLT, which
then enhances the mesospheric reversals in both hemispheres, as a consequence of the increased eastward and westward GW
drag in the SH and NH around 80-100 km. The secondary enhancement of the westward GW drag takes place in the lower
thermosphere, for example, in the SH around $120\,\mathrm{km}$, owing to the enhanced dissipation of the surviving faster GW harmonics
due to increasing molecular viscosity with height, which has been previously discussed.

175        The meridional circulation presented in Fig. 4b shows a strengthening of the mesospheric circulation by $1\text{-}2\,\mathrm{m\,s^{-1}}$, i.e.
a stronger southward wind as well as an enhancement of the lower thermospheric mean northward circulation, which are
primarily driven by the intensification of the mesospheric and lower thermospheric GW momentum deposition. Owing to the
enhanced mesospheric meridional circulation the upward (downward) movement in the polar region in the SH (NH) intensifies,
which leads to a stronger adiabatic cooling (warming). This effect can be seen in the mesospheric temperature differences
(Fig. 4c), which are negative (positive) in the SH (NH) around $80\,\mathrm{km}$ in the polar region with up to $-3\,\mathrm{K}$ ($+5\,\mathrm{K}$). A similar
effect but in the opposite sense happens in the lower thermosphere (around $110\,\mathrm{km}$), where there is a relative adiabatic warming
in the SH polar latitudes but a relative cooling in the NH middle- to high-latitudes. Higher up in the thermosphere ($> 120\,\mathrm{km}$)
the combined effect of the changes in the GW-induced mean meridional circulation and GW heating/cooling lead to a slight
relative cooling of a few $\mathrm{K\,d^{-1}}$ with respect to the control simulation, while the changes in the GW induced heating/cooling are
overall relatively small (Figure 5c), despite the significant increase in the initial source strength. The only exception is the NH
polar thermosphere below $100\,\mathrm{km}$, where the GW heating/cooling characteristic dipole structure with respect to the altitude
shows some intensification.



## 4.2 Modified GW Spectrum: Same Total Flux

We next present the simulation results of EXP3, in which we have increased the number of wave harmonics from 30 to 34,
keeping the maximum phase speed, the total momentum flux, and the peak momentum flux the same as in the benchmark case,
EXP1. For this purpose, the FWHM was reduced to $c_w = 26\,\mathrm{m\,s^{-1}}$. This adjustment has shifted the phase speeds to slightly
larger values in the tail of the spectrum, while significantly decreasing the momentum flux they carry (Fig. 1). The impact of
the spectral changes, primarily in the tail of the spectrum, which is now populated with slightly faster waves but with smaller
wave fluxes can be seen in the zonal wind (Fig. 6a). It shows that the height of the wind reversal is slightly shifted upward
and its magnitude is reduced, indicated by the relative negative/positive zonal wind differences, EXP3-EXP1, around $100\,\mathrm{km}$
in the SH and NH, respectively. Overall, the strengths of the GW momentum flux controls the magnitude of the zonal wind
reversal in the MLT. Higher up in the NH lower thermosphere, the easterlies become weaker by more than $20\,\mathrm{m\,s^{-1}}$ compared
to EXP1. These changes can be explained by the reduction of the momentum flux in the tail of the spectrum in EXP3 relative
to EXP1.

The meridional circulation (Fig. 6b) in the mesosphere as well as in the thermosphere also shows a weakening. The south-
ward wind (northward wind) between $80$ and $100\,\mathrm{km}$ ($100$ and $120\,\mathrm{km}$) is reduced by more than $2\,\mathrm{m\,s^{-1}}$ ($4\,\mathrm{m\,s^{-1}}$). The
weakening of both circulation patterns affects the intensity of the downward and upward movements in the polar region, which
is therefore also less pronounced. This leads to a colder (warmer) winter (summer) mesopause, which can be seen in the nega-
tive (positive) temperature anomalies of $-3\,\mathrm{K}$ ($+3\,\mathrm{K}$) shown in Fig. 6c. In the polar thermosphere, the effect is even stronger
with more than $+5\,\mathrm{K}$ temperature difference. These changes are primarily controlled by the associated changes in the zonal
GW drag, rather than by the changes in the meridional GW drag. The meridional GW drag anomalies are confined to the polar
latitude in the NH MLT (Fig. 7b). Up to $8$-$10\,\mathrm{K\,d^{-1}}$ relative reduction in the resultant GW cooling controls the thermal budget
of the lower thermosphere to a much lesser extent than the dynamical changes.

Between $100$ and $120\,\mathrm{km}$ the zonal GW drag difference (Fig. 7a) shows a strong positive anomaly of more than $+50\,\mathrm{m\,s^{-1}\,d^{-1}}$,
which may be an effect of the vertical shift of the eastward directed zonal GW drag on the SH. While the zonal GW drag anoma-
lies are stronger on the SH, the meridional GW drag anomalies (Fig. 7b) are more pronounced in the NH, especially in the
polar thermosphere. We observe a weakening of the meridional GW drag of more than $20\,\mathrm{m\,s^{-1}\,d^{-1}}$. Thereby, the region of
southward directed GW drag near $120\,\mathrm{km}$ in EXP1 (Fig. 3b) disappears in EXP3 (Fig. 7b). The northward directed GW drag
near $110\,\mathrm{km}$, however, persists. With respect to the GW heating anomalies (Fig. 7c), the induced cooling in the thermosphere
is strongly reduced by more than $10\,\mathrm{K\,d^{-1}}$. Thus, the EXP3 cooling is half the EXP1 cooling.

Note that in EXP2, we had increased the source peak momentum flux by more than a factor of $1.5$. This corresponds to
increasing the momentum flux of each GW harmonic by $50\%$ as well as the total momentum flux by $50\%$. In EXP3, however,
we have kept the total momentum flux constant, as well as the peak momentum flux and the phase speed range, varying only the
FWHM of the spectrum and the number of harmonics. Interestingly, the response of the mesosphere and lower thermosphere
in terms of changes in circulation patterns and temperature distributions are stronger in EXP3 than in EXP2, which emphasizes
the dynamical significance of the faster (nonorographic) GWs for the structure of the mesosphere and lower thermosphere. This





means that uncertainty in the peak momentum flux of the GW spectrum has less impact than the uncertainty in the spectral
shape of the spectrum on the structure of the atmosphere up to the lower thermosphere.

## 5   Relation between Gravity Waves and Terdiurnal Tides

225    The migrating TDT amplitudes and phases in the zonal wind, meridional wind, and temperature are shown in Fig. 8 for the
benchmark simulation EXP1. The amplitudes of all components have a maximum in the lower thermosphere between 120 and
140 km, being larger in the summer SH than winter NH. Ground based radar observations and satellite measurements of the
TDT activity have overall focused on the MLT region between 80 to 110 km and have reported an autumn and winter maximum
of the TDT amplitudes (e.g., Beldon et al., 2006; Jacobi, 2012; Liu et al., 2019; Pancheva et al., 2013; Yue et al., 2013). The

TDT amplitudes simulated by the GCM are also larger in the winter hemisphere than the summer hemisphere in the MLT,
in relatively good agreement with these measurements. Vertical wavelengths in the MLT and above, taken from the vertical
TDT phase gradients in Fig. 8d-f, are on the order of 30 km in the summer hemisphere, but larger in winter. This agrees with
observations (Thayaparan, 1997; Jacobi, 2012).

   The most important dynamical feature in the lower thermosphere in our modeling study are GWs of lower atmospheric

origin parameterized by the whole atmosphere GW parameterization, and therefore they are the most obvious candidate for
tidal modulation at those heights. We next want to examine their influence on the TDT. The terdiurnal components of the GW
parameters presented in Fig. 9 can be used as a proxy for GW-TDT interactions. The amplitudes of terdiurnal GW drag and
heating maximize near 120-130 km, where TDT amplitudes in temperature and wind maximize as well. This indicates that the
TDT in the thermosphere is strongly influenced by the momentum deposition of dissipating GWs. We also present phases of

terdiurnal GW drag and heating in Fig. 9d-f. Similar to the wind and temperature TDT (Fig. 8d-f), the phases of the terdiurnal
GW effects are rather irregular in the mesosphere but become more organized in the lower thermosphere with longer vertical
wavelengths.

   Figures 10a-c show the TDT amplitudes for the simulation EXP2 (contour lines), i.e. the one with increased GW source
flux, and their differences with respect to EXP1 (color shading). Their general structure in wind and temperature is similar

to that of EXP1 (Fig. 8a-c). Differences EXP2-EXP1 amount to $\pm 1$ to $\pm 2\,\mathrm{m\,s^{-1}}$ and K. The terdiurnal signature in GW
parameters (Fig. 10d-f) are mainly increased near their maxima, which can be interpreted as a direct result of the increased
source momentum flux in simulation EXP2. There are also some negative changes between EXP2 and EXP1 but these are
rather irregular and most likely a result of the slightly altered background dynamics described in section 4.1, influencing wave
propagation conditions in general.

Furthermore, there is good agreement between the TDT amplitude changes in the zonal wind (Fig. 10a) and zonal GW drag
(Fig. 10d). For example, the latter one is increased at northern low latitudes at about 120-140 km and decreased at southern
low latitudes at a similar altitude. This pattern is visible in TDT zonal wind amplitude changes, as well. In the meridional com-
ponents (Fig. 10b,e), the positive/negative change patterns also largely agree. In the thermal component (Fig. 10c,f), however,
the TDT temperature amplitude seems to be damped where the difference in terdiurnal GW heating is positive.





Figure 11 is similar to Fig. 10, but refers to the differences between EXP3 and EXP1. This means that differences are based on a GW spectrum with more harmonics in EXP3 than in EXP1, but a smaller momentum flux for each individual harmonic, in particular in the mid-range phase speeds (see Fig. 1). As a result, the sign in the EXP3-EXP1 differences of terdiurnal GW parameters is opposite to that of EXP2-EXP1 differences, i.e. it is negative almost everywhere (see Fig. 11d-f). Accordingly, the TDT wind amplitudes (Fig. 11a,b) are also mostly smaller in the EXP3 simulation compared to EXP1, decreasing by

about $1\text{-}2\,\mathrm{m\,s^{-1}}$ in the area of maximum amplitudes. They partly increase by a similar magnitude, but the decrease dominates, especially in the zonal wind component. Similar to the differences of EXP2-EXP1, the relation between terdiurnal GW heating (Fig. 11f) and TDT temperature amplitude (Fig. 11c) is less clear and we also observe large patches of positive amplitude changes up to $1.5\,\mathrm{K}$ in the temperature component that seem to be corresponding to negative GW heating.

## 6    Summary and Conclusions

We have successfully implemented a whole-atmosphere GW paramterization according to Yiğit et al. (2008) into the mechanistic MUAM GCM. The zonal mean horizontal wind patterns in the middle atmosphere as well as the global temperature distribution is realistically reproduced by the model, and reasonably agree with respect to established climatologies and other GCM predictions. We have successfully simulated the self-consistent direct GW penetration into the thermosphere as has been previously done by other GCMs using the whole atmosphere scheme (Yiğit et al., 2009; Miyoshi and Yiğit, 2019). Compared

to earlier MUAM versions that use a highly tuned linear Lindzen-type GW parameterization for the middle atmosphere (e.g. Jacobi et al., 2015; Lilienthal and Jacobi, 2019; Samtleben et al., 2019), the winter mesospheric jet is enhanced in the present simulations. In the lower thermosphere, the main impact of the whole atmosphere GW parameterization is to drive a vertical/meridional circulation, mainly by the GWs with high phase speeds, which are able to propagate through the MLT wind jets.

We performed two further experiments to investigate the response of the middle and upper atmosphere dynamics to changes of the initial GW spectrum and horizontal momentum flux. In the first experiment (EXP1), we increased the momentum flux at the source level by roughly $50\%$ of the original value, while keeping the spectral shape unchanged. As expected, these modifications result in a stronger GW dissipation at lower levels (near $80\,\mathrm{km}$), connected with a slightly intensified wind reversal in the MLT. This intensifies both the upper mesosphere and lower thermosphere meridional winds, related to stronger adiabatic

warming/cooling of the winter mesosphere/lower thermosphere and reverse effects in the summer hemisphere. However, it turns out that the relatively drastic change of the peak source momentum flux does not influence the global dynamical patterns by such a large degree. One may conclude that an adjustment of this parameter is not crucial for the implementation of this GW parameterization into a GCM like MUAM.

In the final experiment (EXP3), the total momentum flux was kept constant with respect to the benchmark case EXP1,

but the total number of harmonics was increased and the width of the spectrum was decreased, yielding smaller fluxes for the high phase speed tail of the spectrum. In this experiment, the mesospheric wind system was less affected, but the lower thermospheric jets were weakened, connected with cooling/warming of the winter/summer mesosphere, and reversed thermal





effects above. Considering EXP2 and EXP3 with respect to the benchmark case, in general, the lower thermospheric circulation and temperature distributions responded more strongly to the changes in the spectral shape of the GW spectrum than to the
increase in the peak momentum flux. This emphasizes the dynamical significance of the faster GWs for the structure of the lower thermosphere, as these waves are less affected by dissipation and filtering processes in the stratosphere and mesosphere, and thus penetrate into the thermosphere (e.g. Yiğit et al., 2008, 2009; Yiğit and Medvedev, 2017).

We also investigated the effect of the GW source distribution on the amplitudes of the TDT. The simulated latitudinal-vertical distribution of TDT amplitudes, and their vertical wave structure was found to be realistic when compared with observations.
Modifications of the GW source spectrum or total momentum flux mainly influences the TDT in the lower thermosphere, and to a much lesser degree in the upper mesosphere. We found that increasing the GW momentum flux essentially leads to an increased terdiurnal variation in the GW drag and increased amplitudes in the lower thermosphere, which we interpret as a direct result of increased momentum flux of fast GWs. In turn, narrowing of the width of the GW spectrum mostly leads to a reduced terdiurnal variation in the GW drag and lower TDT amplitudes, as a consequence of the reduced momentum
flux of fast GWs in the lower thermosphere. Miyahara and Forbes (1991) already demonstrated in their simulations that an interaction between the DT and GWs can generate a secondary TDT. Recently, GW-tide interactions were again underlined as an important excitation mechanism of the TDT (Lilienthal et al., 2018; Lilienthal and Jacobi, 2019). With respect to DT and SDT amplitudes, Ribstein and Achatz (2016) pointed out that tidal amplitudes are very sensitive to the respective implemented GW parameterization. Thus, it is reasonable that a modified GW drag also strongly influences the TDT amplitudes as has been
shown in our simulations.

Our modeling experiments highlight the importance of taking into account self-consistent propagation of GWs and their dissipation in the mesosphere and lower thermosphere. GWs play a crucial role for the mesospheric and thermospheric circulation patterns and temperature variations as well as for the TDTs. It is noteworthy that small modifications of momentum fluxes of fast GWs have substantial impact on the lower thermosphere mean circulation, and also on tidal amplitudes in particular with
respect to the TDT.

*Code availability.* The MUAM model code is archived on Zenodo and can be freely accessed via http://doi.org/10.5281/zenodo.3628282.

*Author contributions.* FL designed and performed the MUAM model runs together with NS. The first version was drafted by EY (introduction), FL and NS (main part) and CJ (summary). In later versions, EY contributed significantly towards finalizing the main part. All authors discussed the results.

*Competing interests.* The authors declare that they have no conflict of interest.



*Acknowledgements.* Friederike Lilienthal, Nadja Samtleben and Christoph Jacobi acknowledge support by DFG through grants JA836/30-1, JA836/32-1, and JA836/38-1. ERA-Interim data have been provided by ECMWF on https://apps.ecmwf.int/datasets/data/interim-full-moda/levtype=sfc/.





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



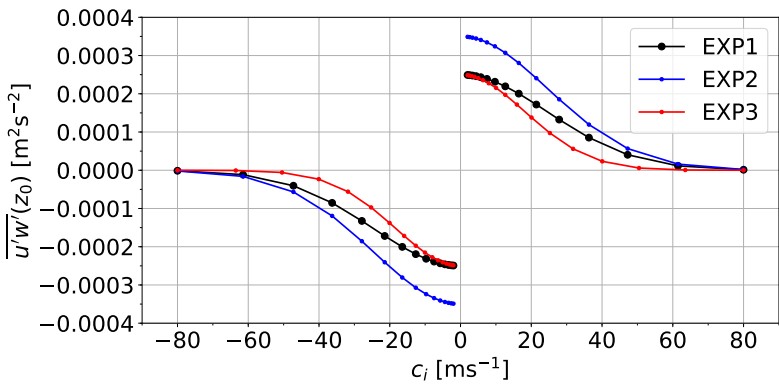

**Figure 1.** Spectra of GW momentum fluxes at the source level of the GW parameterization, $\overline{u'w'}(z_0)$, as a function of horizontal phase speeds $c_i$ for EXP1 (black), EXP2 (blue) and EXP3 (red).

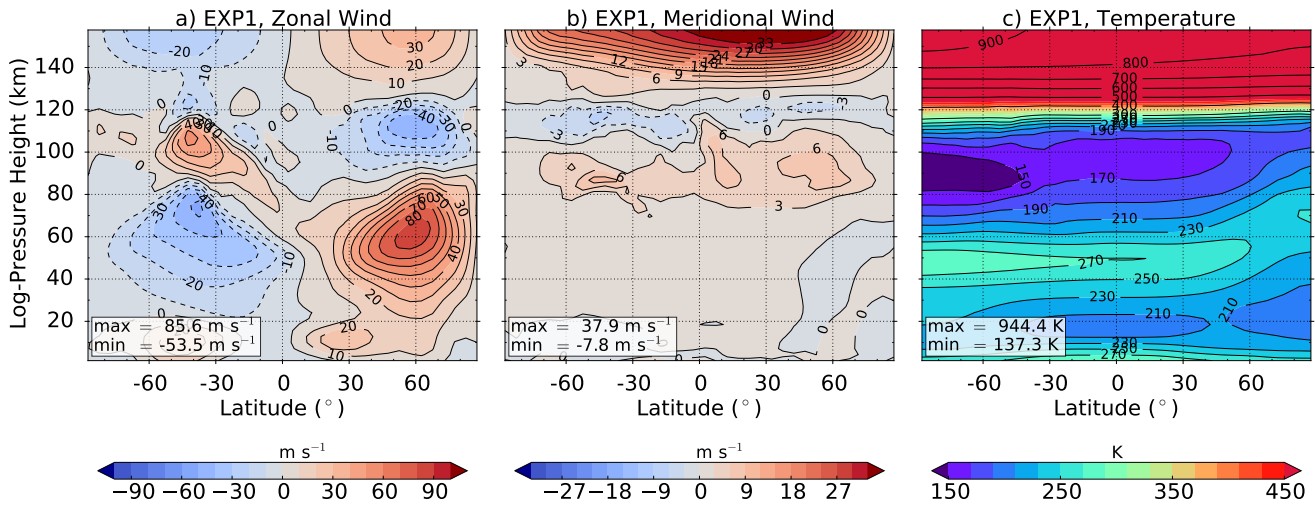

**Figure 2.** Zonal mean zonal wind (a), meridional wind (b) and temperature (c) for EXP1 simulation. Units in (a,b) $\mathrm{m\,s^{-1}}$ and (c) K. Contour lines show intervals of (a) $10\,\mathrm{m\,s^{-1}}$, (b) $3\,\mathrm{m\,s^{-1}}$ and (c) 20 K (for $T < 300\,\mathrm{K}$) and 100 K (for $T \geq 300\,\mathrm{K}$), respectively.





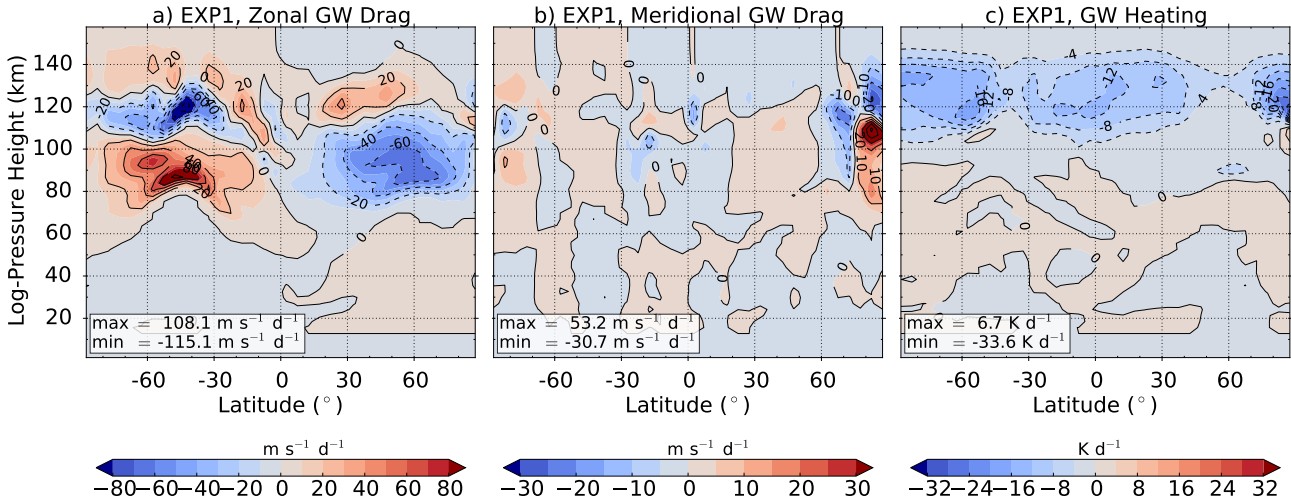

**Figure 3.** Same as Fig. 2 but for zonal GW drag (a), meridional GW drag (b) and heating due to GWs (c). Units in (a,b) $\mathrm{m\,s^{-1}\,d^{-1}}$ and (c) $\mathrm{K\,d^{-1}}$, respectively.

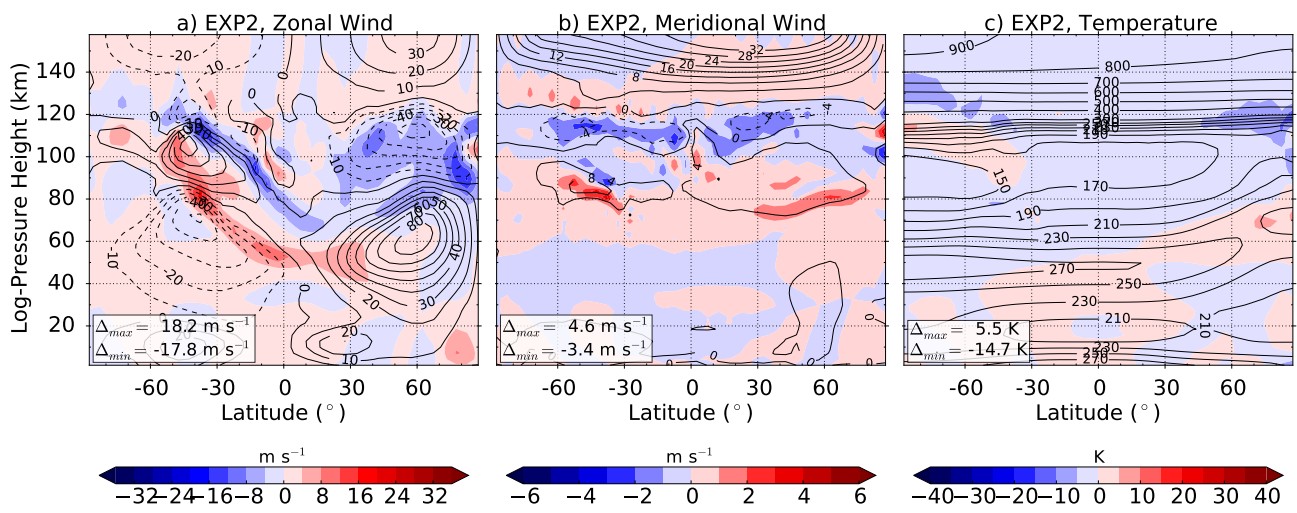

**Figure 4.** Contour lines: Zonal mean (a) zonal wind, (b) meridional wind and (c) temperature for EXP2 simulation in intervals of (a) $10\,\mathrm{m\,s^{-1}}$, (b) $4\,\mathrm{m\,s^{-1}}$ and (c) 20 K below 300 K and 100 K above. Color shading: differences EXP2-EXP1.





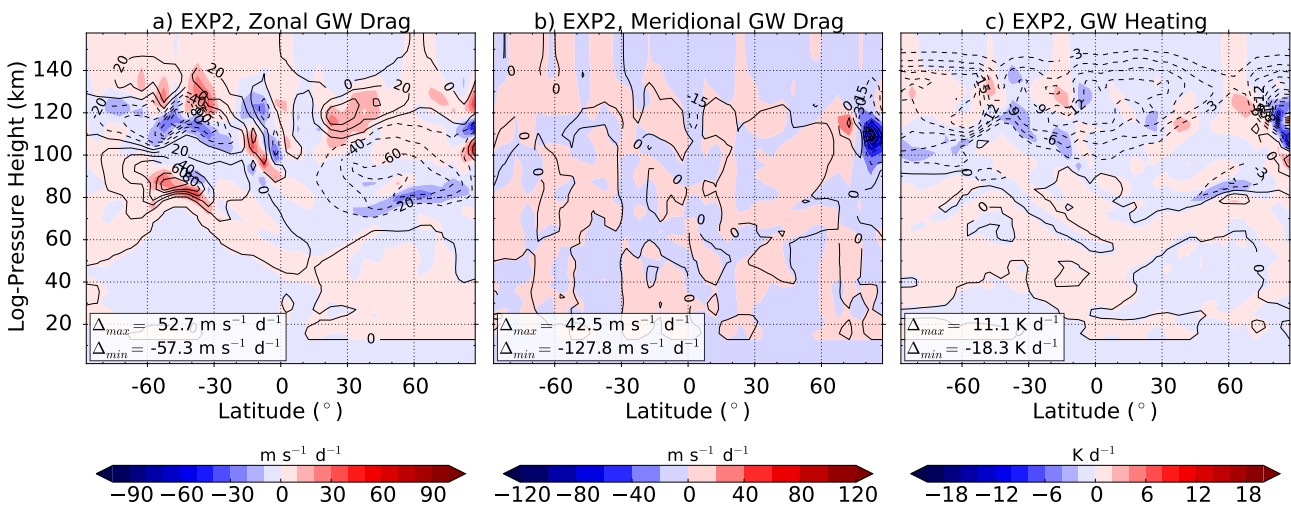

**Figure 5.** Same as Fig. 4 but for (a) zonal GW drag, (b) meridional GW drag and (c) heating due to GWs in intervals of (a) $20\,\mathrm{m\,s^{-1}\,d^{-1}}$, (b) $15\,\mathrm{m\,s^{-1}\,d^{-1}}$ and (c) $3\,\mathrm{K\,d^{-1}}$. Color shading: differences EXP2-EXP1.

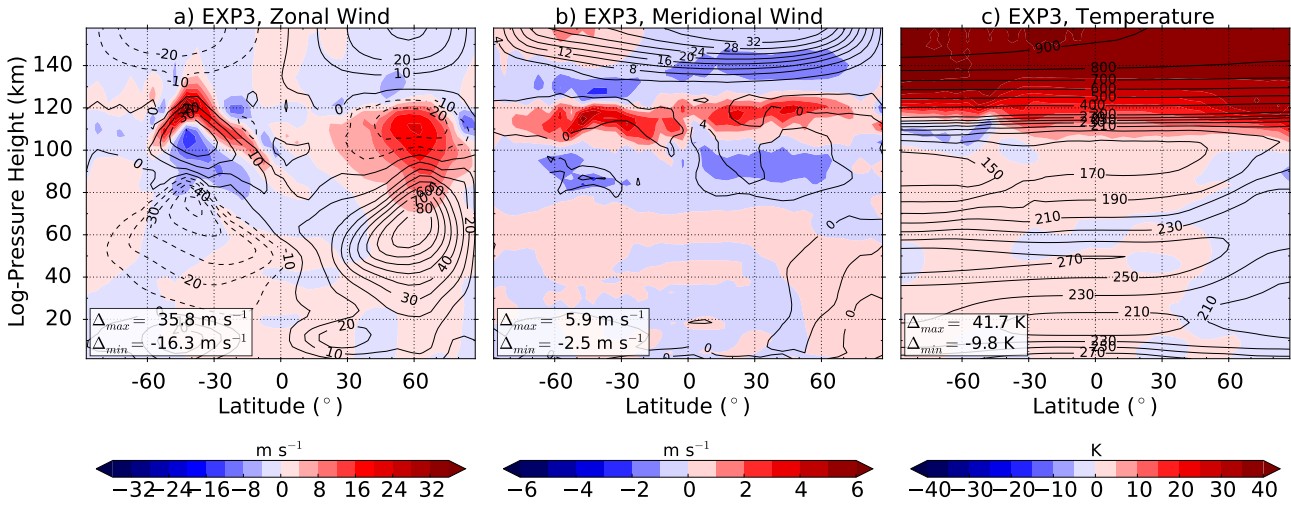

**Figure 6.** Contour lines: Zonal mean (a) zonal wind, (b) meridional wind and (c) temperature for EXP3 simulation in intervals of (a) $10\,\mathrm{m\,s^{-1}}$, (b) $4\,\mathrm{m\,s^{-1}}$ and (c) $20\,\mathrm{K}$ below 300 K and 100 K above. Color shading: differences EXP3-EXP1.





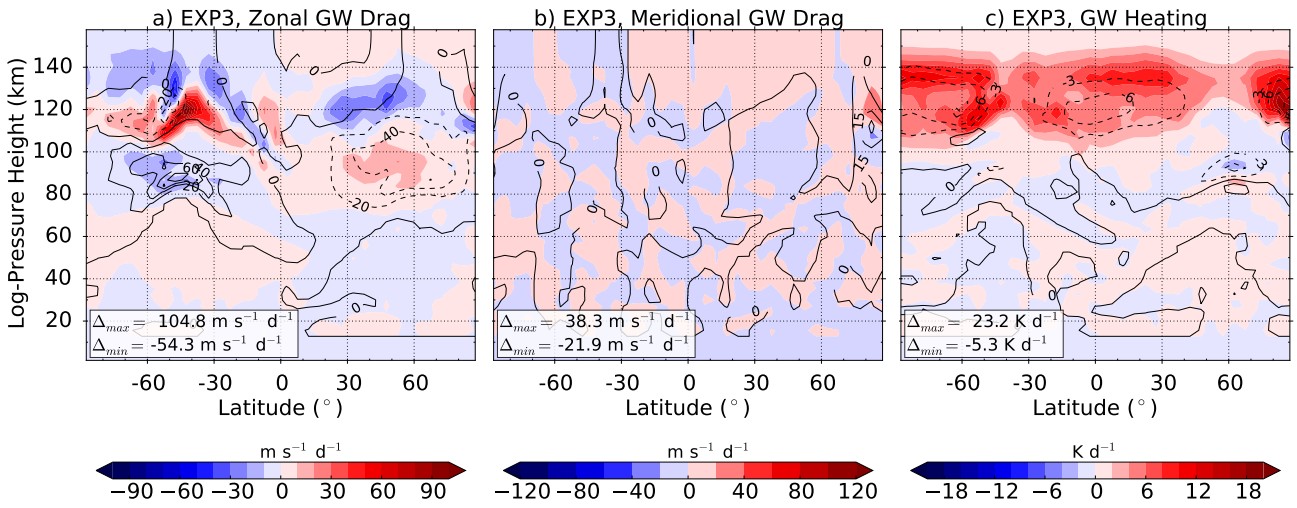

**Figure 7.** Same as Fig. 6 but for (a) zonal GW drag, (b) meridional GW drag and (c) heating due to GWs in intervals of (a) $20\,\mathrm{m\,s^{-1}\,d^{-1}}$, (b) $15\,\mathrm{m\,s^{-1}\,d^{-1}}$ and (c) $3\,\mathrm{K\,d^{-1}}$.



**Figure 8.** Zonal mean TDT amplitudes of (a) zonal wind (in m s$^{-1}$), (b) meridional wind (in m s$^{-1}$) and (c) temperature (in K) for EXP1. (d-f): corresponding TDT phases (in radians).



**Figure 9.** Same as Fig. 8 but for TDT amplitudes of (a) zonal GW drag (in $m\,s^{-1}\,d^{-1}$), (b) meridional GW drag (in $m\,s^{-1}\,d^{-1}$) and (c) heating due to GWs (in $K\,d^{-1}$), and (d-f) corresponding phases (in radians).



**Figure 10.** Zonal mean TDT amplitudes of EXP2 for (a) zonal wind, (b) meridional wind, (c) temperature, (d) zonal GW drag, (e) meridional GW drag and (f) heating due to GWs (black contour lines). Intervals are (a,b) $2\,\mathrm{m\,s^{-1}}$, (c) $2\,\mathrm{K}$, (d) $1.5 \cdot 10^{-4}\mathrm{m\,s^{-1}\,d^{-1}}$, (e) $0.5 \cdot 10^{-4}\mathrm{m\,s^{-1}\,d^{-1}}$ and (f) $0.2 \cdot 10^{-4}\mathrm{K\,d^{-1}}$. Positive/negative differences EXP2-EXP1 ($\Delta$) are shaded in red/blue. Their maximum/minimum values ($\Delta_{max}/\Delta_{min}$) are given in each panel.

**Figure 11.** Same as Fig. 10 but for EXP3 (contour lines) and differences EXP3-EXP1 (color shading).