# Peer review of "Interaction of Small-Scale Gravity Waves with the Terdiurnal Solar Tide in the Mesosphere and Lower Thermosphere"

_Geoscientific Model Development, 2019_

## Referee Comment (RC1) · Nikolai M. Gavrilov (Referee) · 29 Feb 2020

The paper is devoted to the implementation of a parameterization of Gravity wave (GW) effects to the numerical model of the atmospheric general circulation. The authors included a new version of GW spectral parameterization into the Middle and Upper Atmosphere Model (MUAM). The paper contains descriptions of numerical experiments for studying sensitivity of the simulated circulation and tidal amplitudes to GW momentum flux scenarios. The results are new and valuable for the developers of atmospheric dynamical models. The paper corresponds to the scope of the "Geoscientific Model Development" and can be published in this journal after minor modifications.

[Figure]

A few main comments could be made about the manuscript text:

Lines 95 – 100. The authors refer the GW spectrum used previously by Yigit and Medvedev. However, there is no information about reasons of using this particular spectrum. May be it was described in previous papers. However, it would be useful to give short summary of these reasons.

In addition, the authors use spectral function of horizontal speed only. However, for complete GW characterization a second parameter (period or wavelength) is required. How much the GW parameterization is sensitive to changes in periods or wavelengths?

Lines 100 – 105. The authors perform three numerical experiments (labeled as EXP1 – EXP3) using different values of GW momentum fluxes and different spectra of GW phase speed shown in Figure 1. It would be desirable to give a short description and instructions for readers, how considered GW parameters can correspond to different typical meteorological situations.

Please also note the supplement to this comment:
https://www.geosci-model-dev-discuss.net/gmd-2019-339/gmd-2019-339-RC1-supplement.pdf

**Supplement:**

**Review**
of the paper by F. Lilienthal et al. "Interaction of Small-Scale Gravity Waves with the Terdiurnal Solar Tide in the Mesosphere and Lower Thermosphere"

The paper is devoted to the implementation of a parameterization of Gravity wave (GW) effects to the numerical model of the atmospheric general circulation. The authors included a new version of GW spectral parameterization into the Middle and Upper Atmosphere Model (MUAM). The methods and assumptions are valid and clearly described. The results of the model testing are sufficient to support the interpretations and conclusions. The code of the model is included into an open database and it is accessible for any reader. The descriptions are sufficient for practical usage of the model. The paper contains sufficient number of references to previous related papers. The title and abstract reflect the content of the paper. The model name is included in the paper. The paper is clearly structured. The language is enough precise.

The paper contains descriptions of numerical experiments for studying sensitivity of the simulated circulation and tidal amplitudes to GW momentum flux scenarios. The results are new and valuable for the developers of atmospheric dynamical models. The paper corresponds to the scope of the "Geoscientific Model Development" and can be published in this journal after minor modifications.

A few main comments could be made about the manuscript text:

Lines 95 – 100. The authors refer the GW spectrum used previously by Yigit and Medvedev. However, there is no information about reasons of using this spectrum. May be it was described in previous papers. However, it would be useful to give short summary of these reasons.

Also, the authors use spectral function of horizontal speed only. However, for complete GW characterization a second parameter (period or wavelength) is required. How much the GW parameterization is sensitive to changes in periods or wavelengths?

Lines 100 – 105. The authors perform three numerical experiments (labeled as EXP1 – EXP3) using different values of GW momentum fluxes and different spectra of GW phase speed shown in Figure 1. It would be desirable to give a short description and instructions for readers, how considered GW parameters can correspond to different typical meteorological situations.

---

## Referee Comment (RC2) · Anonymous Referee #2 · 4 Mar 2020

Recommendation: major revision

** Summary

The study is devoted to the implementation of a new gravity wave parameterization into a mechanistic circulations model which falls into the field covered by GMD. The particular impact of gravity wave launch spectra on circulation patterns and terdiurnal tides is elucidated, in particular the different role of intensity and width. The response of tides depends on many details of their generation and propagation, which all are influenced, among others, by gravity waves. In order to qualify the influence of gravity wave parameterizations, the old and new scheme should be compared with each other

and be validated against available observations. Such information is pretty sparsely scattered in the text and should be presented with additional figures and text. It is also necessary because this is the first publication on the new gravity wave parameterization in MUAM. Acknowledging this additional work I recommend major revision.

** Major comments

1) Comparison: In the text you mention some changes in dynamics which are due to the change from the former Lindzen-type to the new Medvedev&Klaasen-type gravity wave parameterization. I think, there are some important circulation patterns which can only be properly treated with the new scheme. However, for the setup you mention some simplifications, such as no eddy diffusivity, which should also be further documented. So I suggest to add a detailed comparison between the two schemes.

2) Validation: These different circulation and tidal patterns should be related to available observations and simulations. Some of those are mentioned in the text, but I think the paper requires substantially more information in terms of text and figures. A detailed discussion of relevant publications (for example Becker, 2017 or Liu et al., 2018) in this field is expected.

** Technical comments

L48: Insert "The" before "next..."

L65: "extrem" –> "extreme"

L75: Without "around the world" it is more neutral style.

L79: "due molecular" –> "due to molecular"

L113: Abbreviation for "Southern Hemisphere" was defined already in L109.

L188: I think the narrower spectrum is the key - suggest to write "Modified GW Spectrum: Narrower flux at Source Level"

L192-194: These sentences confuse me: If you make the spectrum narrower, you shift the phase speeds to lower values and increase the momentum flux they carry. All in all, the total momentum flux remains the same. What you write, is the opposite ("shift.. phase speed... to larger values... decreasing the momentum flux they carry")

L239: May be you wish to give a number like "a TDT of 22 m/s is changed by a GWD_TDT of 7.5 m / s / d over 6 hrs by 2.5 m/s which makes up 20 %" or so.

L276: Didnt you want to write "In the additional experiment (EXP2)..."?

** References

Becker, E., 2017: Mean-Flow Effects of Thermal Tides in the Mesosphere and Lower Thermosphere. J. Atmos. Sci. 74, 6: 2043-2063, doi:10.1175/jas-d-16-0194.1.

Liu, H.-L., C. G. Bardeen, B. T. Foster, P. Lauritzen, J. Liu, G. Lu, D. R. Marsh, A. Maute, J. M. McInerney, N. M. Pedatella, L. Qian, A. D. Richmond, R. G. Roble, S. C. Solomon, F. M. Vitt & W. Wang, 2018: Development and Validation of the Whole Atmosphere Community Climate Model With Thermosphere and Ionosphere Extension (WACCM-X 2.0). J. Adv. Model. Earth Syst. 10, 2: 381-402, doi:10.1002/2017ms001232.

---

## Referee Comment (RC3) · Anonymous Referee #3 · 10 Mar 2020

This is a well-written manuscript, but this reviewer is confused about what the paper's focus was supposed to be, and what it turned out to be. Also, the gravity wave spectra used for the numerical experiments did not seem to be rooted in any clearly articulated or compelling physical reasoning.

In the Introduction, please explain what new science that the current paper provides in the context of what is already provided in Lilienthal et al. (2018) and Lilienthal and Jacobi (2019). The titles of those papers sound like they cover the same topics as the current paper.

In the Introduction, it is stated that a "significant amount of work has been conducted
on the relation between GWs, DTs, and SDTs", but then the authors go on to quote some rather old papers by Miyahara and Forbes (1991) and Manson et al. (2002) in the context of providing examples of the "vast majority of the studies" (that) "focus on the MLT region in the context of GW-tide interactions". Are there not more current and comprehensive works focusing on GW-DT and GW-SDT interactions to quote?

This paper spends a lot of its time and effort on the zonal mean circulation and thermal structure, whereas according to the build-up in the Introduction, and the title of the paper, this work ought to be more focused on GW-TDT interactions. Maybe the title needs to be changed.

The changes in GW spectra in experiments EXP1, EXP2 and EXP3 are not very big, and they do no produce very big changes in the TDT. What is the thinking behind the changes in these spectral parameters? What is the physical basis for including only one horizontal wavelength in the spectrum? Aren't the GW with higher momentum fluxes at shorter wavelengths, i.e.,

---

## Short Comment (SC1) · 10 Mar 2020

Dear authors,

in my role as Executive editor of GMD, I would like to bring to your attention our Editorial version 1.2:

https://www.geosci-model-dev.net/12/2215/2019/

This highlights some requirements of papers published in GMD, which is also available on the GMD website in the 'Manuscript Types' section:

http://www.geoscientific-model-development.net/submission/manuscript_types.html

In particular, please note that for your paper, the following requirements have not been met in the Discussions paper:

- "If the model development relates to a single model then the model name and the version number must be included in the title of the paper. If the main intention of an article is to make a general (i.e. model independent) statement about the usefulness of a new development, but the usefulness is shown with the help of one specific model, the model name and version number must be stated in the title. The title could have a form such as, "Title outlining amazing generic advance: a case study with Model XXX (version Y)"."

Therefore please add the name/acronym and version number of the MUAM model in the title and code availability section of your revised article.

Yours,

Astrid Kerkweg

---

## Author Comment (AC1) · 4 Apr 2020

Dear Astrid Kerkweg,

you critisize the missing name and version number of our MUAM model in the title and code availability section. Thank you for this hint. They will be included in the revised version to meet the requirements of GMD.

With kind regards,

Friederike Lilienthal et al.

---

## Author Comment (AC2) · 5 Apr 2020

Dear Prof. Dr. Gavrilov,

we wish to thank you very much for your valuable comments and ideas to help improve our mauscript. Below, we are addressing each of your comments.

1. Lines 95 - 100. The authors refer the GW spectrum used previously by Yiğit and Medvedev. However, there is no information about reasons of using this particular spectrum. May be it was described in previous papers. However, it would be useful to give short summary of these reasons.
The whole atmosphere gravity wave (GW) parameterization has been initially developed in the work by Yiğit et al. (2008), in which various spectral shapes have been tested. However, when the whole atmosphere GW scheme has been implemented into a GCM in the work by Yiğit et al. (2009), we have validated the GW source spectrum and found out that the utilized empirical source spectrum successfully reproduces the large-scale structure of the middle atmosphere dynamics. The chosen momentum flux values in the spectrum are comparable to the observed GW activity in the lower atmosphere. Therefore we use the original GW spectrum as the reference source spectrum with which we have experimented in our current study. Overall, it is important to note that the general principle in choosing a particular spectrum is to optimize the response of the middle atmosphere circulation.

2. In addition, the authors use spectral function of horizontal speed only. However, for complete GW characterization a second parameter (period or wavelength) is required. How much the GW parameterization is sensitive to changes in periods or wavelengths?

The horizontal wavelength of GWs in this parameterization is set to a representative value of 300 km, to which a significant portion of the small-scale GW activity can be statistically attributed to. All three-dimensional GCMs using a GW scheme assume a one-dimensional GW spectrum, often prescribing GWs in terms of horizontal momentum fluxes as a function of horizontal phase speeds. So, while we choose a single representative horizontal wavelength, a broad spectrum of phase speeds are implemented, i.e., 2-80 m/s. Thereby, we adopt a range of GW periods, as for a fixed wavelength, the period of wave is inversely proportional the phase speed. In a realistic atmosphere, the wave period is modulated by the background atmosphere.

The sensitivity of the parameterization with respect to the horizontal wavelength is relatively small, considering the typical ranges of a few hundred kilometers. These comments have been added to the revised version of the manuscript.

3. Lines 100 - 105. The authors perform three numerical experiments (labeled as
EXP1 – EXP3) using different values of GW momentum fluxes and different spectra of GW phase speed shown in Figure 1. It would be desirable to give a short description and instructions for readers, how considered GW parameters can correspond to different typical meteorological situations.

In general our GCM study is not designed to study variable meteorological situations. Rather the purpose of our whole atmosphere scheme is to represent the majority of the nonorographic GWs, including a broad rang of harmonics, in a statistical manner. A direct connection between individual waves and typical meteorological connections cannot be established in our framework. However, we acknowledge that a broad spectrum of gravity waves are generated by a range of weather processes, which our GW spectrum covers to a large extent. Often the observed and the modeled GW spectra are very variable and there are marked uncertainties. Therefore, here we perform a range of modest sensitivity tests within the range of established observations.

---

## Author Comment (AC3) · 5 Apr 2020

Dear anonymous Referee2,

we wish to thank you very much for your valuable comments and ideas to help improve our mauscript. Below, we are addressing each of your comments.

1) Comparison: In the text you mention some changes in dynamics which are due to the change from the former Lindzen-type to the new MedvedevKlaasen-type gravity wave parameterization. I think, there are some important circulation patterns which can only be properly treated with the new scheme. However, for the setup you mention some

simplifications, such as no eddy diffusivity, which should also be further documented. So I suggest to add a detailed comparison between the two schemes.

GW dissipation occurs due to a combination of various dissipation processes, such as eddy viscosity, nonlinear wave-wave interactions, molecular diffusion and thermal conduction, and ion drag (Yiğit et al. 2008, 2009; Yiğit and Medvedev 2010). In the mesosphere and lower thermosphere, the most dominant dissipation mechanism is due to the nonlinear interactions among the different GW harmonics (Yiğit et al. 2008). Eddy viscosity plays a relatively minor role in this context. Also, there is a significant degree of uncertainty in eddy viscosity in the MLT, so we chose to exclude this minor effect in our study. Moreover, Yiğit et al. (2008)'s study extensively compared the nonlinear whole atmosphere scheme to the Lindzen scheme (section 7) and demonstrated the unphysical nature of the linear scheme. Without artificially reducing the GW drag, Lindzen scheme produces very large GW drag, which is rather unrealistic and would destabilize the model. Lindzen scheme works fine provided that an extensive amount of tuning is performed. Therefore, we updated our modeling framework with a state-of-the-art nonlinear whole atmosphere GW parameterization, whose physics and application have been discussed and tested in a number of papers.

We agree with the reviewer that certain features of GWs, and thus circulation patterns, can only be treated with the nonlinear scheme. The vast majority of those properties have been discussed in previous publications cited in our manuscript. Therefore, we have not addressed them here in detail. For example, in comparison with Lindzen scheme, the nonlinear wave-wave interactions in our scheme lead to breaking levels lower in the atmosphere with smaller GW drag, which is more realistic. No artificial tuning factors have been used in our scheme and GW momentum deposition occurs naturally over a range of altitudes and not just at a single breaking level, to name some of the features of the nonlinear scheme and differences to the Lindzen scheme.

2) Validation: These different circulation and tidal patterns should be related to available observations and simulations. Some of those are mentioned in the text, but I think

the paper requires substantially more information in terms of text and figures. A detailed discussion of relevant publications (for example Becker, 2017 or Liu et al., 2018) in this field is expected.

We will extend our discussion and validation. We plan to compare our results, e.g., with model outputs of the WACCM-X, HWM and NRL-MSIS and with URAP data.

Technical Comments We strongly appreciate the formal and technical comments to improve the writing. They will certainly be implemented in the next version.

---

## Author Comment (AC4) · 5 Apr 2020

Dear anonymous Referee3,

we wish to thank you very much for your valuable comments and ideas to help improve our mauscript. Below, we are addressing each of your comments.

1. This is a well-written manuscript, but this reviewer is confused about what the paper's focus was supposed to be, and what it turned out to be. Also, the gravity wave spectra used for the numerical experiments did not seem to be rooted in any clearly articulated or compelling physical reasoning.

We are sorry for the confusion and, based on your review, we conclude that the title of the manuscript needs to be adjusted in order to better represent its content: "Variability of gravity wave effects on the zonal mean circulation and migrating terdiurnal tide as studied with the Middle and Upper Atmosphere Model (MUAM2019) using a whole atmosphere nonlinear gravity wave scheme".

We disagree with the reviewer concerning the gravity wave spectra used for the numerical experiments. The whole atmosphere gravity wave (GW) parameterization has been initially developed in the work by Yiğit et al. (2008), in which various spectral shapes have been tested. However, when the whole atmosphere GW scheme has been implemented into a GCM in the work by Yiğit et al. (2009), we have validated the GW source spectrum and found out that the utilized empirical source spectrum successfully reproduce the large-scale structure of the middle atmosphere dynamics. The chosen momentum flux values are comparable to the observed GW activity in the lower atmosphere. Therefore we use the original GW spectrum as the reference source spectrum with which we have experimented in our current study. While observations provide a better coverage of GW activity nowadays, there are still a certain degree of uncertainties/errors in the observed fluxes. Here we have performed a systematic way of sensitivity tests with the GW spectrum to find out to what properties of the GW spectrum the atmosphere responds. Of course, the uncertainty or variability in the GW source activity is very relevant to the question of GW interactions with the terdiurnal tide: Variation in GW source activity will influence GW effects at higher altitudes, which can impact the terdiurnal tide. A more accurate representation of GW processes in the mesosphere and lower thermosphere can improve our understanding of tidal-GW interactions. Overall, our study of sensitivity tests based on modified GW spectra and the study of GW-terdiurnal interactions are well connected.

2. In the Introduction, please explain what new science that the current paper provides in the context of what is already provided in Lilienthal et al. (2018) and Lilienthal and Jacobi (2019). The titles of those papers sound like they cover the same topics as the

current paper.

The previous papers by Lilienthal et al. (2018) and Lilienthal and Jacobi (2019) mainly focus on the variety and relative importance of forcing mechanisms of the terdiurnal tide. The present manuscript, however, aims to outline the importance of the GW parameterization used, not only with respect to tides but also its impact on the zonal mean circulation. Here, for the first time, we study the GW-TDT interactions with a state-of-the-art whole atmosphere GW parameterization, which gives us a more confident basis to study the influence of GWs on the TDT as the new GW scheme realistically propagates the subgrid-scale GWs through mesosphere into the thermosphere. To highlight this issue, and according to Referee2, we will also include further discussion of the non-linear whole atmosphere parameterization according to Yiğit et al. (2008) compared to the earlier used linear Lindzen-type scheme and compare with other measurements and models.

3. In the Introduction, it is stated that a "significant amount of work has been conducted on the relation between GWs, DTs, and SDTs", but then the authors go on to quote some rather old papers by Miyahara and Forbes (1991) and Manson et al. (2002) in the context of providing examples of the "vast majority of the studies" (that) "focus on the MLT region in the context of GW-tide interactions". Are there not more current and comprehensive works focusing on GW-DT and GW-SDT interactions to quote?

We will add further and more recent publications, here.

4. This paper spends a lot of its time and effort on the zonal mean circulation and thermal structure, whereas according to the build-up in the Introduction, and the title of the paper, this work ought to be more focused on GW-TDT interactions. Maybe the title needs to be changed.

We have expanded the discussion on the TDT and also adjusted the title of the manuscript to reflect the content of the paper (see comment above).

5. The changes in GW spectra in experiments EXP1, EXP2 and EXP3 are not very big, and they do no produce very big changes in the TDT. What is the thinking behind the changes in these spectral parameters?

First, we aim to demonstrate that the whole atmosphere parameterization works very well for a mechanistic global circulation model like MUAM, improving the underlying physics of GW propagation and dissipation. Furthermore, we can show the robustness of this parameterization with respect to different GW phase speed spectra. The sensitivity tests are all within the range of uncertainties of the observed GW parameters in the lower atmosphere. Nevertheless, some differences in the spetra indicate possible interactions with the TDT which supports the present understanding of additional tidal forcing mechanisms, besides the main solar forcing.

6. What is the physical basis for including only one horizontal wavelength in the spectrum? Aren't the GW with higher momentum fluxes at shorter wavelengths, i.e., < 100 km?

Here we used a GW spectrum that has been tested and validated in previous modeling studies cited in our manuscript, demonstrating a realistic mean circulation in the middle atmosphere. GW parameterizations have to be used in GCM to resolve the subgrid-scale waves. These schemes reduce the computational cost, efficiently accounting for the missing GW physics. To maintain high computational efficiency, often a representative horizontal wavelength is used in GW parameterizations in GCMs. In a statistical manner an important portion of the GW activity can be attributed to 300 km horizontal wavelength. Of course it is possible that there are multiple wavelengths present in the atmosphere for a given moment, however, our results in the mesosphere and lower thermosphere are less sensitive to the horizontal wavelength as variations in the wavelength weakly influence GW dissipation compared to other parameters. From the perspective of GW propagation and dissipation the most important two aspects are (1) an accurate representation of physics of GW dissipation and (2) intrinsic phase speed of GWs.

7. Is there a difference between the part of the spectrum that is effective in determining the zonal mean circulation, and the part of the spectrum that interacts efficiently with tides? If so, please discuss in the context of making the choices that you do in the parameters for EXP1, EXP2, EXP3.

At present, we cannot seperate the spectrum into parts that rather effectively change the zonal mean circulation and those that affect the tides. Fast and slow GWs both strongly influence the zonal mean circulation, but at different geographical locations and different heights (as described in the manuscript). As the zonal mean circulation is a very important factor for tidal propagation, the whole spectrum of GWs can also have an impact on the tides.

8. What do Sections 3 and 4 and Figures 2-7 (the bulk of the paper !!) have to do with GW-TDT interactions, which is supposed to be the main focus of this paper?

Sections 3 and 4 will be better addressed in the new title.